# Lipidomics Provides New Insight into Pathogenesis and Therapeutic Targets of the Ischemia—Reperfusion Injury

**DOI:** 10.3390/ijms22062798

**Published:** 2021-03-10

**Authors:** Zoran Todorović, Siniša Đurašević, Maja Stojković, Ilijana Grigorov, Slađan Pavlović, Nebojša Jasnić, Tomislav Tosti, Jelica Bjekić Macut, Christoph Thiemermann, Jelena Đorđević

**Affiliations:** 1School of Medicine, University of Belgrade, 11129 Belgrade, Serbia; maja.stojkovic@med.bg.ac.rs (M.S.); jbjekic@yahoo.com (J.B.M.); 2University Medical Centre “Bežanijska kosa”, 11080 Belgrade, Serbia; 3Faculty of Biology, University of Belgrade, 11000 Belgrade, Serbia; sine@bio.bg.ac.rs (S.Đ.); jasnicn@bio.bg.ac.rs (N.J.); jelenadj@bio.bg.ac.rs (J.Đ.); 4Institute for Biological Research “Siniša Stanković”—National Institute of Republic of Serbia, University of Belgrade, 11000 Belgrade, Serbia; iligri@ibiss.bg.ac.rs (I.G.); sladjan@ibiss.bg.ac.rs (S.P.); 5Faculty of Chemistry, University of Belgrade, 11000 Belgrade, Serbia; tosti@chem.bg.ac.rs; 6Translational Medicine and Therapeutics, William Harvey Research Institute, Barts and The London School of Medicine and Dentistry, Queen Mary University of London, London EC1M 6BQ, UK; c.thiemermann@qmul.ac.uk

**Keywords:** ischemia/reperfusion, lipidomics, kidney, liver

## Abstract

Lipids play an essential role in both tissue protection and damage. Tissue ischemia creates anaerobic conditions in which enzyme inactivation occurs, and reperfusion can initiate oxidative stress that leads to harmful changes in membrane lipids, the formation of aldehydes, and chain damage until cell death. The critical event in such a series of harmful events in the cell is the unwanted accumulation of fatty acids that leads to lipotoxicity. Lipid analysis provides additional insight into the pathogenesis of ischemia/reperfusion (I/R) disorders and reveals new targets for drug action. The profile of changes in the composition of fatty acids in the cell, as well as the time course of these changes, indicate both the mechanism of damage and new therapeutic possibilities. A therapeutic approach to reperfusion lipotoxicity involves attenuation of fatty acids overload, i.e., their transport to adipose tissue and/or inhibition of the adverse effects of fatty acids on cell damage and death. The latter option involves using PPAR agonists and drugs that modulate the transport of fatty acids via carnitine into the interior of the mitochondria or the redirection of long-chain fatty acids to peroxisomes.

## 1. Introduction

Ischemia-reperfusion (I/R) injury is still a significant medical problem. Despite new therapeutic possibilities, acute reperfusion damage to the kidneys, liver, heart, and other organs is accompanied by high mortality. Renal ischemia and reperfusion are among the most common causes of prerenal acute kidney injury (AKI), while liver I/R is frequently associated with organ transplantation, liver resection, and trauma [1].

Ischemia impairs cell metabolism, while reperfusion may aggravate the injury caused by ischemia alone. In brief, during ischemia, there is a reduced ATP synthesis in a mainly anaerobic environment and impaired plasmalemmal ion-exchange with calcium overload. During reperfusion, mitochondrial damage and swelling occur, calcium overload further increases, and an imbalance between prooxidants and antioxidants results in oxidative stress. Ultimately, cell death may be due to apoptosis, mitoptosis, necrosis, necroptosis, ferroptosis, and/or autophagy [2,3].

Mitochondria have a central role in the chain of metabolic events that accompany I/R injury, with the calcium homeostasis disturbance as the main one. The Ca^2+^ homeostasis in a variety of mammalian organs, including the heart [4], the kidney [5], and the liver [6], is regulated by Na^+^/Ca^2+^ electrogenic exchanger (NCX). NCX is able to function in both forward (Ca^2+^ efflux) and reverse (Ca^2+^ influx) modes [7]. Under normal circumstances, the NSC operates in the efflux mode, in which Ca^2+^ is expelled from the cell, reducing its intracellular concentration. However, during ischemia, the NCX switches into influx mode, in which calcium is transferred into the cell, increasing its intracellular concentration [8]. The resulting Ca^2+^ intracellular overload then activates the opening of the mitochondrial inner membranes permeability transition pore (mPTP) [9]. As a result, water enters the mitochondria, leading to the cristae swelling, membrane rupture, and consequent cell necrosis (Figure 1).

During reperfusion, NCX returns to the influx mode of operation. However, the mPTP opening is now activated by the radical oxygen species (ROS), whose generation is strongly increased by reperfusion (Figure 1). Cellular death due to necrosis results in an uncontrolled release of Damage-Associated Molecular Pattern Molecules, such as high mobility group Box 1 protein (HMGB1), into the extracellular space [10], and decreased protein expression of nuclear factor erythroid 2-related factor 2 (Nrf2) [11,12]. The Nrf2 is one of the most important inducible transcription factors that protects against ischemic damage by enhancement of the endogenous antioxidant system, such as copper-zinc superoxide dismutase (CuZnSOD), manganese superoxide dismutase (MnSOD), catalase (CAT), glutathione peroxidase (GSH-Px), glutathione (GSH), glutathione reductase (GR), glutathione S-transferase (GST), and haem oxygenase 1 (HO-1) [13]. Finally, the I/R-caused release of the inflammatory cytokines further propagates inflammation, spreading it across the neighboring organs, a phenomenon known as the “organ crosstalk” [14]. It should be noted that during I/R, necrotic and apoptotic cell death occur simultaneously because of the mitochondrial outer membrane rupture that leads to the release of numerous pro-apoptotic factors [15], such as Bax protein [11,12] or cytochrome C [16].

Experimental approaches for the treatment of ischemia-reperfusion injury have focused on factors in the early and late stages of development of this disorder. In the kidneys, many substances were effective in preclinical models, but the translation of these interventions to the clinic was not successful. This can be explained by the inadequacy of preclinical models to fully reflect clinical conditions (including the lack of co-morbidities), a lack of understanding of the pathophysiology of the I/R injury, or the need to combine multiple interventions [17,18]. Briefly, antioxidants, nitric oxide synthase (NOS) inhibitors, peroxisome proliferator-activated receptor (PPAR) agonists, poly(adenosine 5′-diphosphate ribose) polymerase (PARP) inhibitors, poly(ADP-ribose) glycohydrolase (PARG) inhibitors, erythropoietin, statins, chloroquine, and other substances were tried with variable success [19,20,21,22]. They were all focused on different targets such as NO and related enzymes, ROS, proinflammatory transcription factors, apoptosis regulators, calcium overload, etc. (for details, see Ref. [2]).

## 2. Lipidomics

The determination of the lipidome profile is often called lipidomics. The lipidome represents all the small molecules metabolomes with a mass lower than 1500 in the system [23]. Lipids are involved in many biological processes in organisms due to their hydrophobic part, such as building biological membranes, keeping energy for further consumption, and playing an essential role in cell signaling. Production and lipids concentrations must be strictly controlled because any lipid metabolism dysregulation can lead to disease [24]. Hence, nowadays, the analysis of the lipid profile is a rapidly developing field.

Based on their hydrophobic moiety, lipids are divided into eight categories: fatty acids (FAs), prenols, sterols, glycerophospholipids, glycerolipids, sphingolipids, polyketides, and saccharolipids (Table 1) [25].

### 2.1. Lipid Transport Across Cell Membranes

In particular, there is an increased accumulation of free FAs in certain types of tissue damage with subsequent cell death (“lipotoxicity”), which is why incorporation of FAs into the TG pool serves as a vital cell-protective mechanism [26]. The FAs have a vital function in physiology, being directly synthesized in the cytosol (in situ), released from intracellular metabolic processes (e.g., hydrolysis of triglycerides, TGs, and phosphatidylcholine, PC), or obtained from extracellular sources through CD36-mediated uptake [27]. The enzymes involved in TG synthesis, such as acyl-coenzyme A:diacylglycerol acyltransferases (DGATs), are present in various tissues, for example, kidneys and liver [28,29].

Besides the internalization of lipoprotein particles (LDL, chylomicron remnants, and IDL) by endocytosis in the presence of LDL receptors and their degradation in lysosomes, various other proteins participate in the transmembrane transport of lipids. Such transport may play a significant role in the pathogenesis of IR injury.

CD36 (fatty acid translocase), a member of the class B2 scavenger receptor family of cell surface proteins, is found on blood cells (platelets, erythrocytes, monocytes), adipocytes, hepatocytes, as well as cells of the myocardium, spleen, and renal tubule epithelium. Ligands for CD36 are proteins and lipids, and the latter includes oxidized LDL particles, long-chain fatty acids, phospholipids, and others. CD36s are involved in fatty acid metabolism, atherosclerosis, and other processes. Their role in long-chain fatty acid uptake is particularly important. In the liver, CD36s are directly linked to fatty acid metabolism, and in the kidneys, they are involved in the uptake of advanced oxidation protein products and the development of lipototoxicity. In the heart, more than 70% of ATP is formed from fatty acids, and CD36 mediates 70% of the intake of fatty acids into the cardiac cells [30]. The number of CD36s on the cell membrane of cardiomyocytes decreases in ischemia and remains low during reperfusion. Reducing the number of CD36s in ischemia reduces the transport of long-chain fatty acids into cells and their breakdown in mitochondria. Simultaneously, the number and activity of GLUT4 transporters for glucose and anaerobic glucose metabolism to pyruvate increases. Pyruvate is further metabolized in the mitochondria to form ATP. Simultaneously, the cytoplasm’s pH decreases due to proton accumulation, which interferes with the transport of CD36 from the endosome back to the cell membrane. The reduced number of CD36 in the cell membrane prevents the lipotoxic accumulation of fatty acids in the fat chains and enables the survival of cells in ischemia. During reperfusion, under aerobic conditions, anaerobic glycolysis and proton accumulation decrease. However, protons accumulated during ischemia further interfere with the transport of CD36 back to the cell membrane. Simultaneously, the production of ATP by aerobic oxidation of long-chain fatty acids in mitochondria increases now, and the metabolism of pyruvate, as a product of glycolysis, declines.

The fatty acid transport proteins (FATPs) are part of the family of the solute carrier 27 (Slc27) proteins which also have an essential place in the transport of exogenous fatty acids [31,32]. It should be emphasized that FATPs play the role of a gateway in such transport, regardless of whether they are located on the cell membrane or intracellularly, at the junction of the membranes with the endoplasmic reticulum. Therefore, some representatives of this group of transport proteins, such as FATP-1, -2, and -4, are suitable targets for drug action. By the way, six members of this group of proteins, FATP1–6, have been identified in mammals, and some of them are very long chain acyl CoA synthetases.

The family of fatty acid-binding proteins (FABP) has 12 members (“lipid-binding chaperones”) and plays a role in the intracellular transport of fatty acids, eicosanoids, retinoids, and other lipophilic compounds, their trafficking, and signaling [33,34]. Their presence in the liver, intestine, peripheral nervous system, skeletal and heart muscle, adipocytes, skin, and brain has been confirmed. Besides the transport of fat, FABPs are involved in vasculogenesis, cell differentiation, and pathogenesis of various metabolic disorders [35]. Their role in transporting lipid compounds to PPAR receptors may be associated with brain IR injury [35]. Additionally, pharmacological inhibition of FABP4 may protect the kidneys from rhabdomyolysis-induced acute kidney injury (AKI) [36]. Namely, BMS309403, a selective inhibitor of FABP4, reduced glycerol-induced renal tubule damage, alleviated endoplasmic reticulum stress in the murine model of this AKI, decreased serum creatinine levels, and expressed proinflammatory cytokine expression. Further, the FABP4 gene induces hepatocyte hypoxia, which sensitizes liver cells in a hepatic IR injury model [37].

Efferocytosis can also be a vital process in transporting lipid substances and mitigating IR injury consequences [38]. This is a specific process that participates in removing apoptotic cells (AC), reducing the possibility of further secondary necrosis and inflammation development. It differs from classical phagocytosis. It takes place through several phases: AC finding, their binding, internalization, and degradation. Efferocytosis reduces inflammation by increasing the production of anti-inflammatory cytokines and reducing the synthesis of proinflammatory molecules. For example, sterols that reach efferocytes stimulate PPAR-gamma, PPAR-delta, and liver X receptor-α, leading to further stimulation of the release of anti-inflammatory IL10 and TGF-beta and differentiation of T cells that suppress inflammation. Such a defense mechanism removes the excess cholesterol that occurs during phagocytosis of apoptotic cells and can be cytotoxic to efferocytes. In other words, AC-derived cholesterol can be esterified under the influence of acyl-CoA cholesterol acyltransferase to cholesterol esters deposited as neutral fat droplets or subject to efflux. AC-derived fatty acids can be broken down by oxidation in mitochondria and further increase the expression of anti-inflammatory IL-10 via sirtuin-1. Finally, binding of phosphatidylserine (PS) of AC membrane to efferocytes leads to upregulation of glucose transporter 1 (GLUT1) and its incorporation into the cell membrane.

The process of FAs oxidation occurs in mitochondria and peroxisomes [39]. FAs can pass through the membrane of mitochondria or peroxisomes only via the carnitine cycle. First, the binding of acyl-CoA to carnitine is catalyzed by carnitine palmitoyl-transferase 1, and the product, long-chain acyl-carnitine (LCAC), subsequently passes through the membranes of mitochondria and peroxisomes via carnitine-acyl-carnitine translocase. Then, free carnitine is regenerated within mitochondria under the influence of carnitine palmitoyl-transferase 2, while the end product, acyl-CoA, undergoes beta-oxidation within mitochondria and peroxisomes in the presence of FAD and NAD. Very-long-chain FAs are metabolized in peroxisomes. Essential regulators of cell lipid metabolism are AMP-activated protein kinase (AMPK) and peroxisome proliferator-activated receptor-α (PPARα) receptors. AMPKs are well-known energy sensors. On the other hand, PPARs form heterodimers with retinoid X receptors (RXR) and serve as ligand-activated transcription factors that play significant roles in the metabolism of lipids and inflammation.

### 2.2. The Role of Fatty Acids in Metabolic Pathways

The essential FAs play a critical role in metabolism, where the key point is the ratio of the polyunsaturated n3 and 6 [40]. Recent research highlights the importance of mono- and polyunsaturated FAs. For example, arachidonic acid (AA) is involved in metabolic pathways as an “ancestor” of the numerous lipid mediators, for example, eicosanoids [41]. The importance of concentration of AA (C20:4n-6), dihomo-γ-linolenic acid (DGLA; C20:3n-6), eicosapentaenoic acid (EPA; C20:5n-3), and docosahexaenoic acid (DHA; C22:6n-3) in the liver reperfusion was emphasized by Kirac et al. They analyzed liver tissue specimens and observed that the ratio of AA/DHA was significantly increased while AA/EPA remained the same [42]. The neuroprotective role of free fatty acids, such as AA and docosahexaenoic acid (DHA; 22:6), in brain I/R injury in rats was the main reason for their rapid accumulation, as shown by Adibhtala et al. [43]. In the same study, the authors concluded that increasing the concentration of ceramide inhibits mitochondrial electron transport, which leads to cell apoptosis. Montero-Bullón et al. showed that I/R and starvation increase the concentration of saturated fatty acids such as C16:0, C18:0, and polyunsaturated C20:4 and C22:6, while C18:2 decreases [41]. Further, the influence of the phospholipid profile on the acute myocardial infarction was analyzed. They observed the increase of the phospholipids bearing FAs, such as phosphatidylcholine and phosphatidylethanolamine compounds PC(18:0–22:6), PE(16:0–22:6), and PE(18:0–22:6). That conclusion was additionally confirmed in the study of Zheng et al., which pointed out the importance of PC and lysophosphatidylcholine (LPC) in brain ischemia [44]. The principal component analysis of this study emphasized the role of the monotonic relationship between the levels of PC(16:0/16:0) and LPC(16:0) and their increment in concentration. De la Monte et al. investigated the white matter cerebral lipid profile in I/R in sheep. They characterized alterations in cerebral white matter lipid profiles in an established foetal sheep model [45]. They observed that changes in CL, PC, phosphatidylinositol monomannoside, sphingomyelin, sulfatide, and ambiguous or unidentified lipids occur mainly after 48 h of I/R (I/R-48) and normalized or suppressed at I/R-72. ROS also increase the levels of oxidative phosphatidylcholines (OxPCs) in the I/R injury in the heart [46]. OxPCs may trigger apoptosis through phosphorylation of p38 mitogen-activated protein kinase (p38MAPK).

In the serum of patients who underwent myocardial infarction, 16 fatty acids were isolated as biomarkers that enable early monitoring and have diagnostic value [47]. After acute cardiac ischemia, serum levels of these fatty acids increase, indicating inhibition of their beta-oxidation (it is the dominant source of ATP in the myocardium under normal conditions). Then, during reperfusion, fatty acids are taken into the myocardium, and their beta-oxidation intensifies. The accumulation of fatty acids in the heart muscle leads to its damage with inhibition of contractility and the appearance of dysrhythmias, as well as increased oxygen consumption without a simultaneous increase in myocardial work.

### 2.3. Lipids as Signaling Molecules

Particular attention should be paid to lipid signaling. It is known that lipids target proteins by modulating the most important processes in the cell. At the same time, they are not deposited but are synthesized de novo, easily passing through the membranes.

One group of the essential lipid signaling molecules are ceramides (“lipid hub”). Ceramides are a family of lipid molecules consisting of sphingosine backbone and fatty acid residues. The fatty acids in ceramide, saturated or mono-unsaturated, have 14 to 26 C chains in length. Ceramides are widely present in the cell membrane, forming sphingomyelin with phosphocholine. Due to their structural characteristics, they could be found in particular cell membrane regions called rafts. Besides the structural role, they also participate in different processes such as cell differentiation and apoptosis. Ceramides are part of the lipotoxicity cascade, whose intracellular concentration increases due to various processes, such as increased synthesis from palmitoyl-CoA and serine, recycling of complex sphingolipids, dephosphorylation of ceramide-1-phosphate, or increased degradation/hydrolysis of sphingomyelin. The latter process may occur in the different phases of the IR injury. For example, early activation of neutral sphingomyelinases in the IR-injured cardiomyocytes in the presence of FAN protein (factor associated with neutral sphingomyelinases, FAN) gives rise to the increased release of ceramide and subsequent apoptosis. Further increase in the ceramide levels in cardiomyocytes occurs during reperfusion due to decreased ceramidase activity [48]. The concentration of ceramide increases in renal IR injury as well [49]. Additionally, ceramide may induce apoptosis of renal tubular cells [50]. A balance between sphingosine-1 phosphate and ceramide (anti- and pro-apoptotic signal, respectively) is essential for such a process. Ceramide stimulates pro-apoptotic Bcl-2 proteins and increases the mitochondrial outer membrane’s permeability, with further formation of reactive oxygen species, cytochrome C release, and activation of effector caspases.

In addition to the other lipid signaling molecules, such as sphingolipid second messengers, second messengers from phosphatidylinositol, or activators of G-protein coupled- and nuclear receptors, exosomes that transmit lipid signals between cells should be mentioned, both in physiological and pathological conditions (for example, ischemia-reperfusion injury, cancer, and heart failure). Exosomes, microvesicles, and apoptotic bodies belong to extracellular vesicles that transmit various signals between cells [51]. The former is released by fusion with the cell membrane and contain lipids in their membrane and intravesical milieu. For example, ceramide, cholesterol, or sphingomyelin are enriched in exosomes relative to the cells from which they originate. Exosome has been investigated during the previous decade, both for diagnostic (biomarkers) and therapeutic purposes. Further research should focus on the role of exosomes in organ crosstalk, which occurs during I/R injury.

## 3. Lipid Metabolism in I/R Injury

Lipids are among the major targets of ROS in oxidative stress [52]. Both free radicals and non-radical ROS (for example, superoxide anion, hydroxyl radical, and hydrogen peroxide, respectively) impair various cell molecules in oxidative stress. Along with ROS, reactive nitrogen species, RNS (e.g., peroxynitrite and nitrogen dioxide) may also affect cellular molecules in reperfusion-related oxidative stress. Activated macrophages and neutrophils release both ROS and RNS. The main intracellular sources of ROS are mitochondria and peroxisomes. Namely, mitochondrial complexes I and III release superoxide anions, which are converted to hydrogen peroxide by superoxide dismutase, and subsequently to highly toxic hydroxyl free radical in the presence of ferrous iron (the Fenton reaction).

There is ample evidence of adverse accumulation of FAs in renal I/R injury, which ultimately may give rise to lipotoxicity [27,46]. The term “lipotoxicity” refers to the condition with an unwanted accumulation of lipids in non-adipose tissues that are incapable of metabolizing them. The full mechanism of lipotoxicity remains to be clarified, but certain steps have already been explained. For example, during ischemia and reperfusion, there is decreased FA beta-oxidation and increased phospholipid hydrolysis, fatty acid uptake, and lipid synthesis. The latter could serve as an early buffer mechanism against fatty acid overload. The accumulation of non-toxic cholesterol and TGs and FAs, as well as diacylglycerol and ceramide (downstream metabolites of unsuccessful esterification or breakdown of complex lipids), was detected in renal I/R injury [53,54]. Of note, even subtle changes in lipid content, structure, function, or location in the cell may significantly impact cell homeostasis [55].

ROS produced in oxidative stress cause the release of lipid radicals from membrane TGs and PCs. The chain reaction begins with the separation of hydrogen from lipids, which gives rise to the lipid radical release and their subsequent oxidation into lipid peroxy-radicals (LOO-) [55]. These radicals are rapidly oxidized with lipid peroxidation of acyl chains. The reaction spreads in a chain with a further restructuring of membrane PCs and trapping of lipid radicals. Peroxidized acyl chains are further cleaved (Hock cleavage) to α-, β-polyunsaturated lipid aldehydes, leaving shortened acyl chains of parent TFs and PCs, which disrupt membrane structure and alter its permeability. Polyunsaturated lipid aldehydes are highly cytotoxic and can modify cellular proteins and nucleic acids, which may lead to mitochondrial dysfunction, the unfolded protein response (UPR), endoplasmic reticulum (ER)-stress, and apoptosis. The type of lipid aldehydes depends on the local milieu, i.e., the species n3 and n6 polyunsaturated FA species that dominate in the tissue affected by oxidative stress. There are different aldehyde-containing oxidized lipid products such as malondialdehyde, acrolein, 4-hydroxyhexenal (HHE), and 4-hydroxynonenal (HNE) [56]. In the liver, HHE predominates among reactive lipid aldehydes. 4-HNE has a significant role in cell damage, having a genotoxic effect, and acting as a secondary free radical messenger. It modulates the level of transcription factors involved in cell protection, such as Nrf2, activating protein-1 (AP-1), nuclear factor kappa-light-chain-enhancer of activated B cells (NF-κB), and peroxisome proliferator-activated receptors (PPAR). We would particularly emphasize the role of lipid hydroperoxides in cell damage because they are much more stable substances than reactive oxygen and nitrogen species.

Cardiolipin (Calcutta antigen) or 1,3-bis(sn-3′-phosphatidyl)-sn-glycerol) is an important component of the inner mitochondrial membrane that accounts for ~20% of its total lipid content. Cardiolipin serves as a stabilizer of mitochondrial cristae and is vital for the integrity of the respiratory chain. On the other hand, it is susceptible to lipid peroxidation and formation of HNE, which is an essential part of the intrinsic (mitochondrial) pathway of apoptosis. Peroxidation of cardiolipin by hydrogen peroxide triggers the release of cytochrome C and its transport to the outer mitochondrial membrane, which initiates apoptosis [57]. Further, oxidized cardiolipin itself migrates to the outer mitochondrial membrane, where it triggers the formation of mitochondrial voltage-dependent anion channel through the interaction with Bax, apoptosis regulator, and a member of Bcl2.

Reactive lipid aldehydes are very stable and easily diffuse inside and outside the cell. On the other hand, they easily react with proteins and nucleic acids. The reaction between lipid aldehydes and proteins is called protein carbonylation. It may cause either loss-of-function modification of certain proteins, for example, enzymes of glycolysis, or a change in protein function and their interactions. In addition, carbonylation may inhibit protein degradation and can be used as a biomarker of oxidative stress. In particular, such modification of proteins within the endoplasmic reticulum (ER) may trigger ER stress and the accumulation of unfolded proteins. Antioxidant enzymes such as glutathione S-transferase inactivate lipid peroxidation products and prevent protein carbonylation.

Lipid peroxidation is closely related to ferroptosis as a type of cell death distinct from apoptosis, necrosis, etc. [58]. Depletion of glutathione or inactivation of glutathione peroxidase 4 triggers ferroptosis. Neither caspase-3-induced cleavage of poly (ADP ribose) polymerase 1 nor the release of cytochrome c from mitochondria are detected. Instead, it is characterized by an excessive peroxidation of polyunsaturated fatty acids, including membrane PCs. Ferroptosis plays a key role in hepatic and renal I/R injury. Surprisingly, Shimada et al. [59] reported that statin-mediated inhibition of HMG-CoA reductase actually facilitates ferroptosis, which is in contrast to many studies showing protective effects of simvastatin in the acute renal I/R injury [22,60]. Such a discrepancy can be explained by the fact that the pleiotropic effects of a single dose of statins initiate other protective mechanisms compared to their lipid-lowering action.

## 4. Therapeutic Approach to I/R Injury-Related Lipotoxicity

Erpicum et al. [27] suggested three possible strategies aimed at preventing lipotoxicity in renal I/R injury:Reduction of the lipid overload in I/R injured tissues by increasing their catabolism and/or extraction;Transportation of lipids towards the adipose tissue;Blockade of the main pathways of FA-induced cell death.

PPARs are members of the superfamily of nuclear receptors with thyroid, steroid, vitamin D3, and retinoid receptors. PPARs regulate lipid metabolism in different ways [61]. For example, PPARα are predominantly involved in FA metabolism, and they can decrease lipid levels in tissues. Their gene targets are, for example, enzymes of the beta-oxidation pathway (e.g., acyl-CoA oxidase, thiolase), sterol 12-hydroxylase (CYP8B1), and FA translocase (FAT/CD36). PARβ/δ are involved in FA oxidation (target gene—FAT/CD36), while PPARγ participate in adipogenesis and energy balance, and they influence genes involved in lipid uptake, metabolism, and efflux.

Sivarajah et al. reported beneficial effects of PPARα agonists clofibrate and WY14643 in the rat model of renal I/R injury almost two decades ago [62]. We detected all three members of the PPAR receptors superfamily in the rat kidneys: PPARα, PPARγ, and PPARβ/δ. Acute renal I/R injury induced only a downregulation of the former subtype, while acute pre-treatment with both PPARα agonists attenuated the I/R-induced glomerular dysfunction, tubular injury, renal histopathology severity score, and intercellular adhesion molecule-1 (ICAM-1) expression. Important findings of Zhou et al. [63] showed that natural polyacetylene glycosides inhibit lipid uptake, biosynthesis, and accumulation in hypoxic renal tubular cells in vitro and attenuate the acute kidney injury in the in vivo model of renal I/R in mice via increased expression of PPARα.

Alvarez-Guardia et al. [64] confirmed that PPARβ/δ receptor activation might attenuate lipid-induced inflammatory pathways. Thiemermann and co-workers demonstrated the beneficial effects of PPARβ/δ agonists in different models of I/R injury [65,66,67]. Similar was found for PPARγ agonists rosiglitazone and ciglitazone in the rat model of renal I/R injury [68]. Another therapeutic approach could be the restoration of tissue carnitine. Propionyl-L-carnitine is a short-chain acyl derivative of L-carnitine that may protect the kidney against I/R injury, as shown by Mister et al. two decades ago [69]. However, meldonium has convincingly shown protective effects in the in vivo models of renal and liver I/R injury in rats [11,12].

## 5. Meldonium as the Therapeutic Approach in I/R Injury-Related Lipotoxicity

Meldonium is an anti-ischemic drug clinically used to treat myocardial and cerebral ischemia [70]. Meldonium inhibits the gamma-butyrobetaine dioxygenase, a final enzyme in carnitine biosynthesis, and the carnitine palmitoyltransferase-1, an inner mitochondria membrane enzyme that catalyzes the transfer of the acyl group from coenzyme-A to carnitine. As a result, long-chain fatty acids transport from cytosol into mitochondria is reduced and redirected to peroxisomes, in which the same process is carried out by ABC proteins, insensitive to meldonium [71]. In peroxisomes, long-chain fatty acids are metabolized to medium- and short-chain acylcarnitines for further oxidation in mitochondria, which prevents the mitochondrial accumulation of toxic long-chain intermediates [72] and reduces the mitochondrial ROS formation whose ratio asymptotically increases with the FAs length [73]. To be precise, the main sites of the mitochondrial ROS production are electron-donating centers of Complexes I and III in the electron transport chain: once reaching a highly reduced state, they are prone to an electron’s “leak” that causes consequent superoxide anion radical production. The redox status of those centers depends on the rate of chain electron flow, whose decrease leads to the centers’ higher reduced state [74]. The catabolic intermediates and the by-product of long-chain FAs are known to be able to slow-down the chain electron flow rate [75], and to disturb the maintenances of mitochondrial redox balance and mitigation of ROS formation [76], thus increasing mitochondrial ROS production. In this way, meldonium decreases the risk of long-chain fatty acids metabolism mediated mitochondrial injury and shifts energy production from fatty acids oxidation to less oxygen-demanding glycolysis, which is more favorable under ischemic conditions.

### 5.1. Meldonium as the Therapeutic Approach in I/R-Mediated Heart Injury

It has been shown that pathological acylcarnitine dysregulation in myocardial cells can be bypassed with meldonium treatment [70]. A reduction in acylcarnitine negatively affects the endoplasmic reticulum and sarcolemma, including the activity of Ca^2+^ and Na^+^-K^+^ ATPase [77]. Thus, meldonium is able to improve myocardial contractility, hexokinase I activity, and cellular glucose oxidation and ATP ratio through activation of AMP-activated protein kinase, an enzyme that restores ATP level [78,79,80].

In line with the above, meldonium proved to be successful in the clinical treatment of various heart disorders, such as early post-myocardial infarction period [81] or chronic heart failure in patients with [82] or without type 2 diabetes mellitus [83]. Similar results were obtained in animal models also. For example, meldonium was able to reduce arrhythmia duration and ventricular fibrillation frequency in rats [84] and decrease blood glucose levels and increase glycaemic metabolism in the mice heart [85]. In addition, a significant reduction of the infarct size, directly linked to a reduction of cardiac l-carnitine and FA pools [86], was shown in the heart of both normal [87,88] and diabetic rats [89]. However, the fact that meldonium had no effects on hemodynamic parameters proved that the anti-infarction effects were related to the metabolic regulation and not to the cardiac workload changes [86,87].

### 5.2. Meldonium as the Therapeutic Approach in I/R-Mediated Brain Injury

Meldonium was shown to exhibit beneficial effects in different animal models of neuroinflammation [90], neurodegeneration [91,92], neuronal apoptosis [93], endothelin 1-induced ischemic stroke [94], streptozocin-induced neuropathic pain [95], memory impairment [96], or schizophrenia [97]. In animal models of the cerebral I/R, meldonium improved functional recovery [98] and increased tolerance against I/R-induced brain injury through mitochondrial uncoupling [99]. Besides its antioxidative effects based on the decrease in mitochondrial long-chain FA accumulation, it has been postulated that meldonium, being structurally similar to L-carnitine, may mimic some of its biological effects (“false carnitine”) [90], such as direct ROS scavenging [100]. It has been shown that in diabetic patients with acute lacunar stroke and dyscirculatory encephalopathy (DEP) meldonium increases blood serum lipoproteins resistance to peroxidation [101].

Neuroprotective effects of meldonium were confirmed in a clinical trial also [70]. Vinnichuk et al. showed that meldonium improves cerebral hemodynamic in patients with stroke and post-ischemic cerebral reperfusion [102]. It seems that the mechanism of meldonium neuronal protection differs somehow from that in the heart, in which, as we have stated above, meldonium anti-infarction effects were not related to the cardiac workload but the metabolic regulation [86,87]. In another trial with the patients with chronic cerebrovascular ischemia, meldonium exerted a positive effect on neurological symptoms, hemodynamic, electrophysiological, and neuropsychological characteristics of the patients [103]. In the trial by Abeuov et al., meldonium decreased the incidence of headaches, dizziness, vestibular dysfunction, and insomnia in patients with dyscirculatory encephalopathy (DEP), and improved memory, attention, and cognition [104]. Finally, it has also been reported that mildronate accelerates regression of brain lesions [105], possibly by modulating brain oxygen consumption [106].

### 5.3. Meldonium as the Therapeutic Approach in I/R-Mediated Hepatic Injury

Acute ischemia/reperfusion (I/R) liver injury is a clinical condition challenging to treat. There are several possible therapeutic options for I/R liver injury, such as organ preservation, inhibition of ROS formation, or immune system activation modulation [107]. While the organ preservation strategy showed the most promising results [108], the use of TNF inhibitors [109], corticosteroids [110], or free-radical scavengers such as N-acetyl cysteine [111], has been mostly unsuccessful or with limited success. For these reasons, we investigated the protective effect of a four-week meldonium pre-treatment in a model of rat hepatic I/R.

In the liver of rats subjected to I/R injury, meldonium exerted anti-inflammatory and antioxidant action, reducing markers of apoptosis and necrosis and increasing antioxidative protection [12]. For example, meldonium decreased both liver and serum levels of HMGB1, previously increased by I/R. This is important since HMGB1 is able to further propagate inflammation through activation of the NF-κB pathway [112]. Accordingly, the hepatic expression of phosphorylated NF-κB p65, previously increased by I/R, was decreased by concurrent meldonium use [12]. Similarly, the hepatic Bax/Bcl2 ratio, previously increased by 50% by I/R, was reduced back to the control level by meldonium. The changes in Bax/Bcl2 ratio indicate meldonium anti-apoptotic effects—it has been suggested that overexpression of anti-apoptotic Bcl2 can block both apoptosis and necrosis [113] and protect ischemic tissue against I/R-induced oxidative stress [114].

In contrast to NF-κB, the hepatic expression of phosphorylated p-Nrf2, previously decreased by I/R, was increased by meldonium use. As a result, the hepatic antioxidative defence, previously significantly weakened by I/R (i.e., 30–50% decrease in hepatic CuZnSOD, MnSOD, and GSH-Px activity, and a 50% and a 2.5-fold decrease in the hepatic HO-1 and GSH content), was restored by concurrent meldonium use (i.e., 50–100% increase in hepatic CuZnSOD, MnSOD, and CAT activity, and a 2.5-fold increase in the GSH-Px activity and GSH content) [12].

In mitochondria, where long-chain FA membrane transport is mediated by carnitine palmitoyl-transferase 1, in peroxisomes, the same process is carried out by ABC proteins, insensitive to meldonium. Our results showed that hepatic I/R gives a uniform rise in tissue concentration of all investigated FAs [12], which indicates its harmful effects due to lipotoxicity. Blockade of mitochondrial FAs overload by meldonium leads to a decrease in hepatic content of all fatty acids, from myristic to the cervonic and erucic acid (C 14:0, and C 22:6/C22:1, respectively). In conditions of reduced oxygen flow during ischemia, tissue damage is less pronounced when ATP is created by aerobic oxidation of glucose instead of the beta-oxidation process of fatty acids.

### 5.4. Meldonium as the Therapeutic Approach in I/R-Mediated Renal Injury

Similar to liver I/R, the acute renal I/R is a challenging condition in clinical medicine due to mortality as high as 50% and lack of efficient pharmacotherapy [20,115]. For these reasons, we investigated the protective effect of a meldonium under the same experimental design as in the case of liver, i.e., using four-week meldonium pre-treatment in a dose of 300 mg/kg b.m./day. This dose was chosen because it is similar to the clinical human use [77,116] and within the range of previously studied animal doses of meldonium [117], where it proved to be both highly effective and safe.

The protective effects of meldonium in renal I/R-mediated injury share many of its effects in hepatic I/R injury but also exerts some differences (Figure 2). For example, while I/R increased the Bax/Bcl2 ratio by 2.7-fold, meldonium reduced this increase by 35% [11]. This finding indicates the reinforcement of pro-apoptotic events during renal I/R and its effective reduction by meldonium. On the other hand, an increase in serum and kidney levels of HMGB1 in animals subjected only to I/R, and their subsequent decrease in animals subjected to both I/R and meldonium, indicates reinforcement of necrotic events during renal I/R and its effective reduction by meldonium. As was the case with the liver, meldonium gives a 2.6-fold rise in renal expression of Nrf2, causing a 1.2-fold increase in renal HO-1 expression.

Unlike the liver, changes in renal Nrf2 expression were not accompanied by appropriate changes in the antioxidative defense system. For example, I/R did not change CuZnSOD, GSH-Px, GR, and GST activities in comparison to controls, while CuZnSOD and GSH-Px activities were even reduced as a result of concurrent meldonium treatment. A possible explanation could be the fact that the total Nrf2 transcriptional activity depends not only on its expression but also on the availability of binding partners, the competition and/or cooperation with other activators and repressors, and crosstalk with other signaling pathways [118].

However, the most prominent difference between liver and kidney was seen in lipidomics. We simultaneously assessed the renal concentrations of 13 FAs: myristic (tetradecanoic, C14:0), pentadecylic (pentadecanoic, C15:0), palmitic (hexadecenoic, C16:0), palmitoleic (cis-Δ9 hexadecenoic, C16:1), margaric (heptadecanoic, C17:0), stearic (octadecanoic, C18:0), oleic/elaidic (cis and trans Δ9-octadecenoic, C18:1 c + t), linoleic/linolelaidic (cis and trans Δ9,12-octadecenoic, C18:2 c + t), behenic (docosanoic, C22:0), cervonic (cis,cis-Δ13,16-docosadienoic, C22:6), and erucic (cis,cis,cis,cis,cis,cis-Δ4,7,10,13,16,19-docosahexaenoic, C22:1); the latter FA is an early sign of renal injury [11]. Meldonium attenuated the I/R-related increase in the concentrations of C22:0 and C22:6 + C22:1 but did not exert the same in sham-operated animals.

The principal component analysis revealed that the sum of C22:6 and C22:1 exhibited the highest positive value, whereas palmitic acid (C 16:0) exhibited the highest negative value. Palmitate plays a vital role in the synthesis of ceramide. According to our results, there was a discrepancy between palmitate changes in the liver and kidney as a result of I/R and meldonium treatment. I/R increases the palmitate content in the liver, which concurrent meldonium use abolishes, i.e., reduces [12]. We assume that this changes the content of ceramide and its harmful effects on the level of mitochondria and cell membranes, which lead to cell death. On the other hand, the opposite changes were detected in the kidneys—a decrease in the palmitate content in the I/R group and an increase in its content after applying meldonium [11]. The mechanism of these changes remains to be clarified, but such a difference clearly indicates a tissue-specific response to both I/R and meldonium.

## 6. Conclusions

Research of pathogenesis and therapy of I/R disorders should inevitably consider lipid metabolism. Lipotoxicity poses a challenge to the analysis of pathogenesis and new therapeutic options to reduce cell death and organ damage. Among the potential drugs, agents that prevent cell damage with fatty acids and their unwanted accumulation, i.e., increase their removal and degradation, have a particular place. Future research in this field should focus on agents that could stop the chain reaction of spreading the effects of lipid peroxidation at the membrane level and prevent excessive accumulation of long-chain fatty acids in cells, i.e., ensure their proper catabolism in mitochondria and peroxisomes. Such substances should act on several targets simultaneously or be combined. A shift in therapy is not possible without monitoring new markers in the complex milieu of I/R injury in the cell. Lipidomics combined with proteomics and genomics opens up the possibility of elucidating the metabolic changes that occur during I/R damage and finding new targets for drug action.

## Figures and Tables

**Figure 1 ijms-22-02798-f001:**
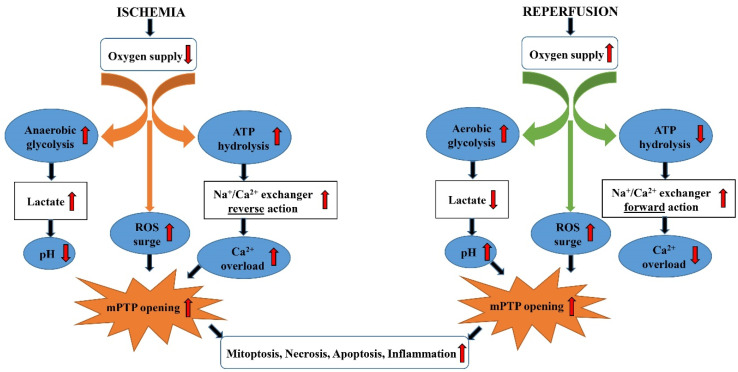
The chain of metabolic events that accompany I/R injury. Na^+^/Ca^2+^—Na^+^/Ca^2+^ electrogenic exchanger (NCX); ROS—reactive oxygen species; mPTP—mitochondrial inner membranes permeability transition pores.

**Figure 2 ijms-22-02798-f002:**
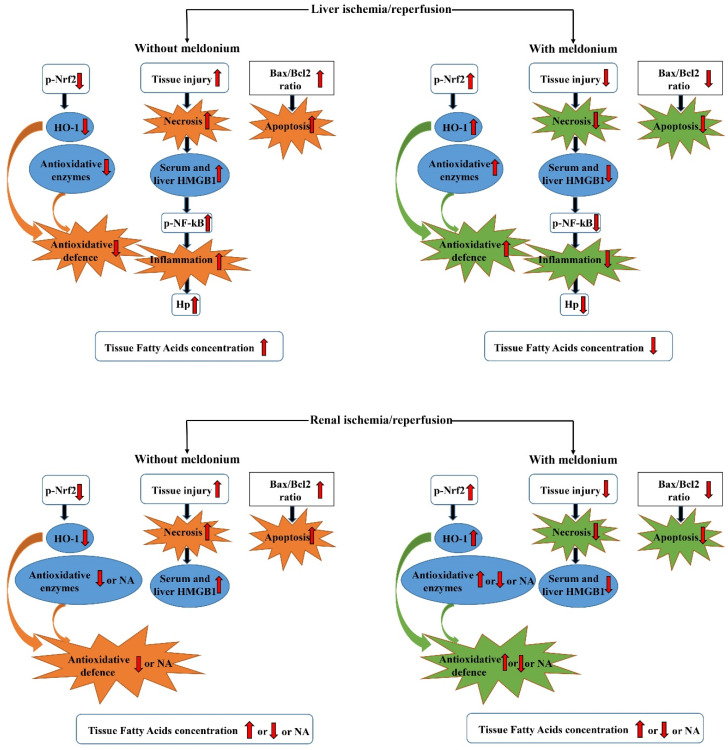
The summarized effects of meldonium on the hepatic and renal ischemia/reperfusion. p-Nrf2—phosphorylated nuclear factor erythroid 2-related factor 2; HO-1—haem oxygenase; HMGB1—high mobility group Box 1 protein; p-NF-kB p65—phosphorylated nuclear factor kappa-light-chain-enhancer of activated B cells; BAX—pro-apoptotic member of Bcl2 gene family; Bcl2—anti-apoptotic member of Bcl2 gene family. Upward pointing arrow (↑) indicates an increase, downward pointing arrow (↓) indicates a decrease, and NA indicates no changes.

**Table 1 ijms-22-02798-t001:** Lipid classification based on Lipid Maps Structure Database.

Lipid Categories	
**01. Fatty Acyls [FA]** [FA01] Fatty Acids and Conjugates [FA02] Octadecanoids [FA03] Eicosanoids [FA04] Docosanoids [FA05] Fatty alcohols [FA06] Fatty aldehydes [FA07] Fatty esters [FA08] Fatty amides [FA09] Fatty nitriles [FA10] Fatty ethers [FA11] Hydrocarbons [FA12] Oxygenated hydrocarbons [FA13] Fatty acyl glycosides [FA00] Other Fatty Acyls **02. Glycerolipids [GL]** [GL01] Monoradylglycerols [GL02] Diradylglycerols [GL03] Triradylglycerols [GL04] Glycosylmonoradylglycerols [GL05] Glycosyldiradylglycerols [GL00] Other Glycerolipids **03. Glycerophospholipids [GP]** [GP01] Glycerophosphocholines [GP02] Glycerophosphoethanolamines [GP03] Glycerophosphoserines [GP04] Glycerophosphoglycerols [GP05] Glycerophosphoglycerophosphates [GP06] Glycerophosphoinositols [GP07] Glycerophosphoinositol monophosphates [GP08] Glycerophosphoinositol bisphosphates [GP09] Glycerophosphoinositol trisphosphates [GP10] Glycerophosphates [GP11] Glyceropyrophosphates [GP12] Glycerophosphoglycerophosphoglycerols [GP13] CDP-Glycerols [GP14] Glycosylglycerophospholipids [GP15] Glycerophosphoinositolglycans [GP16] Glycerophosphonocholines [GP17] Glycerophosphonoethanolamines [GP18] Di-glycerol tetraether phospholipids [GP19] Glycerol-nonitol tetraether phospholipids [GP20] Oxidized glycerophospholipids [GP00] Other Glycerophospholipids	**04. Sphingolipids [SP]** [SP01] Sphingoid bases [SP02] Ceramides [SP03] Phosphosphingolipids [SP04] Phosphonosphingolipids [SP05] Neutral glycosphingolipids [SP06] Acidic glycosphingolipids [SP07] Basic glycosphingolipids [SP08] Amphoteric glycosphingolipids [SP09] Arsenosphingolipids [SP00] Other Sphingolipids **05. Sterol Lipids [ST]** [ST01] Sterols [ST02] Steroids [ST03] Secosteroids [ST04] Bile acids and derivatives [ST05] Steroid conjugates [ST00] Other Sterol lipids **06. Prenol Lipids [PR]** [PR01] Isoprenoids [PR02] Quinones and hydroquinones [PR03] Polyprenols [PR04] Hopanoids [PR00] Other Prenol lipids **07. Saccharolipids [SL]** [SL01] Acylaminosugars [SL02] Acylaminosugar glycans [SL03] Acyltrehaloses [SL04] Acyltrehalose glycans [SL05] Other acyl sugars [SL00] Other Saccharolipids **08. Polyketides [PK]** [PK01] Linear polyketides [PK02] Halogenated acetogenins [PK03] Annonaceae acetogenins [PK04] Macrolides and lactone polyketides [PK05] Ansamycins and related polyketides [PK06] Polyenes [PK07] Linear tetracyclines [PK08] Angucyclines [PK09] Polyether antibiotics [PK10] Aflatoxins and related substances [PK11] Cytochalasins [PK12] Flavonoids [PK13] Aromatic polyketides [PK14] Non-ribosomal peptide/polyketide hybrids [PK15] Phenolic lipids [PK00] Other Polyketides

## Data Availability

Not applicable.

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
