# Peer review of "Lipidomics Provides New Insight into Pathogenesis and Therapeutic Targets of the Ischemia—Reperfusion Injury"

_ijms, 2021, doi:10.3390/ijms22062798_

Round 1

Reviewer 1 Report

Dear Authors,

The authors wrote a review focusing on the importance of the lipids roles in alleviating the injuries due to Ischemia. They provide a list and complete information of different lipids that can be used as treatment. The main targeted organs are the liver and the kidneys, in this review. The review is well written but I have some comments:

1) The authors should have a paragraph reviewing Meldonium or as an introduction about the choice of this compound because it is used to treat heart failure, that could cause brain ischemia. Brain ischemia should also be commented.

2) The authors should explain why they decide to focus on liver and kidneys, and not others organs?

Author Response

Dear referees,

Thank you for the kind and rapid reply. We are grateful to the referees who provided us with very useful comments and did our best to comply with the suggestions. In the revised version of the manuscript, all changes are highlighted in yellow.

Sincerely yours,

Prof. Zoran Todorović, corresponding author

REFEREE #1

1. The authors should have a paragraph reviewing Meldonium or as an introduction about the choice of this compound because it is used to treat heart failure, that could cause brain ischemia. Brain ischemia should also be commented.

We agree with the remark. That is why we have introduced a completely new chapter that is entirely devoted to Meldonium (Lines 326-345). In it, in front of the part related to the liver and the kidney, there are two new parts related to the heart (Lines 346-360) and the brain (Lines 361-382).

2. The authors should explain why they decide to focus on liver and kidneys, and not others organs.

With a new chapter (explanation in the previous question), we believe that we have covered the most common experimental I/R models in which meldonium was used. Also, in the liver (Lines 384-390) and the kidney (Lines 416-421) section, we explained the reasons why we dealt with these I/R models in our experiments.

Reviewer 2 Report

The manuscript of Todorovic et al. (entitled The role of lipidomics in the ischemia/reperfusion injury) reviews the impact of lipidomic approaches in a better understanding of the pathophysiology of ischemia/reperfusion injuries in diverse tissues.

The subject would be of interest for the scientific community, and the present manuscript provides a solid and comprehensive summary of the recent literature, including the own papers of the authors.

The points are clearly presented, the conclusion is correct, but there are some minor issues for criticism that should be addressed in a revised version of the manuscript.

Major point: none

Minor points:

  1. The title of the paper is not correct, since it implies a role of lipidomics in something. Lipidomics has no role in the ischemia/reperfusion injury, since lipidomics is one of the omics technologies. Lipidomics has rather a significant impact in the elucidation of the cellular/molecular pathophysiology of ischemia/reperfusion injuries in diverse tissues.
  2. The fatty acid uptake from extracellular sources is described insufficiently (lines 102-103), since CD36 is not the exclusive fatty acid translocase in the plasma membrane and its expression is not ubiquitious. In addition to CD36, fatty acid transport proteins (FATP1-6/SLC27A1-6) are extensively involved in the cellular uptake of non-esterified fatty acids (NEFAs) transported mainly by albumin. Furthermore, internalization of lipoproteins and lipoprotein-derived fatty acids, and further fatty acids from efferocytosis are also important exogenous sources of fatty acids.
  3. The text body generates a spurious impression that ceramide is one lipid species (lines 163, 165, 362). Instead, ceramide is a group of lipids containing a sphingosine backbone and different fatty acid residues. Different N-acyl-sphingosines have distinct biological functions, but the authors rely mostly on C16:0 and C18:1 ceramides in their discussion.

Author Response

Dear referees,

Thank you for the kind and rapid reply. We are grateful to the referees who provided us with very useful comments and did our best to comply with the suggestions. In the revised version of the manuscript, all changes are highlighted in yellow.

Sincerely yours,

Prof. Zoran Todorović, corresponding author

REFEREE #2

1. The title of the paper is not correct, since it implies a role of lipidomics in something. Lipidomics has no role in the ischemia/reperfusion injury, since lipidomics is one of the omics technologies. Lipidomics has rather a significant impact in the elucidation of the cellular/molecular pathophysiology of ischemia/reperfusion injuries in diverse tissues.

We agree with the remark. That is why we changed the title of the paper into “Lipidomics provides new insight into pathogenesis and therapeutic targets of the ischemia-reperfusion injury”, which we hope is acceptable.

2. The fatty acid uptake from extracellular sources is described insufficiently (lines 102-103), since CD36 is not the exclusive fatty acid translocase in the plasma membrane and its expression is not ubiquitous. In addition to CD36, fatty acid transport proteins (FATP1-6/SLC27A1-6) are extensively involved in the cellular uptake of non-esterified fatty acids (NEFAs) transported mainly by albumin Furthermore, internalization of lipoproteins and lipoprotein-derived fatty acids, and further fatty acids from efferocytosis are also important exogenous sources of fatty acids.

We agree with the remark. That is why we have introduced a new text (Lines 99-157), which deals with the issues you mentioned.

3. The text body generates a spurious impression that ceramide is one lipid species (lines 163, 165, 362) Instead ceramide is a group of lipids containing a sphingosine backbone and different fatty acid residues Different N-acyl-sphingosines have distinct biological! functions, but the authors rely mostly on C16:0 and C18:1 ceramides in their discussion.

We agree with the remark. That is why we have introduced a new text (Lines 204-221), which deals with the issues you mentioned.

Reviewer 3 Report

The authors present a sort of mini-review/perspective article about the impact of lipids in ischemia/reperfusion damage in the liver and kidney. I absolutely agree that lipidomics represents a promising approach to better understand the pathophysiology and the substrate of degenerative diseases, but also in cancers, with promising insights from a translational point of view. 

However, despite these premises, I think that the current article has different flaws: for example, the title should better elucidate that the analysis (a sort of summa of several previous works of the authors) regards liver and kidney. Nothing about the possible impact of brain damage?

The whole article structure is, in some parts, slightly confusing; i.e., is table 1 really necessary? Maybe the reference should be enough

Author Response

Dear referees,

Thank you for the kind and rapid reply. We are grateful to the referees who provided us with very useful comments and did our best to comply with the suggestions. In the revised version of the manuscript, all changes are highlighted in yellow.

Sincerely yours,

Prof. Zoran Todorović, corresponding author

REFEREE #3

1. However, despite these premises, I think that the current article has different flaws. For example, the title should better elucidate that the analysis (a sort of summa of several previous works of the authors) regards liver and kidney. Nothing about the possible impact of brain damage.

We agree with the remark. That is why we have introduced a completely new chapter that is entirely devoted to meldonium (Lines 326-345). In it, in front of the part related to the liver and the kidney, there are two new parts related to the heart (Lines 346-360) and the brain (Lines 361-382).

2. The whole article structure is, in some parts, slightly confusing, i.e., is Table 1 really necessary? Maybe the reference should be enough.

We believe that it is important to put such tables in the review papers since they are more precise than the reference and give the reader the possibility of more convenient navigation concerning the classes of lipids mentioned in the text.

Round 2

Reviewer 1 Report

Dear authors,

The authors answered to all my questions and comments. I accept the manuscript as it is.

Sincerely

Author Response

Dear referee,
Thank you for your valuable comments that have greatly improved our manuscript.
Sincerely,
Zoran Todorovic
(corresponding author)

Reviewer 3 Report

The manuscript has been greatly revised; the main limitation is now represented by the lenght of the section 2 (Lipidomics): i think that some subheading should greatly increase the readers experience

Author Response

Dear referee,
Thank you for your helpful comments that have considerably improved our manuscript. In particular, we have introduced three subheadings in section 2 (Lipidomics).
Sincerely
Zoran Todorovic
(corresponding author)

Round 3

Reviewer 3 Report

I think that the manuscript is now suitable for publication